Presence of an ultra-small microbiome in fermented cabbages

Lee Hae-Won 1 2
Yoon So-Ra 1
Dang Yun-Mi 1
Kang Miran 3
Lee Kwangho 4
Ha Ji-Hyung hajee@wikim.re.kr 1
Bae Jin-Woo baejw@khu.ac.kr 2
1 Hygienic Safety ⋅ Materials Research Group, World Institute of Kimchi , Gwangju , Republic of Korea
2 Department of Biology and Department of Life and Nanopharmaceutical Sciences, Kyung Hee University , Seoul , Republic of Korea
3 Practical Technology Research Group, World Institute of Kimchi , Gwangju , Republic of Korea
4 Center for Research Facilities, Chonnam National University , Gwangju , Republic of Korea
Kumar Ravinder
Electronic publication date: 2023 Jul 17
Publication date: 2023
Volume: 11
Electronic Location ID: e15680
Received 2023 Feb 14; Accepted 2023 Jun 13
Copyright: ©2023 Lee et al.
Copyright year: 2023
Copyright holder: Lee et al.
License: This is an open access article distributed under the terms of the Creative Commons Attribution License, which permits unrestricted use, distribution, reproduction and adaptation in any medium and for any purpose provided that it is properly attributed. For attribution, the original author(s), title, publication source (PeerJ) and either DOI or URL of the article must be cited.
License URL: https://creativecommons.org/licenses/by/4.0/

Keywords: Single molecule real time sequencing, Tangential flow filtration, Ultra-small microbiome, Kimchi, Sauerkraut, Suancai, Fermented cabbage, Ulramicrobacteria, TFF, USM

Funding: National Research Foundation of Korea NRF-2019R1F1A1061368 World Institute of Kimchi KE2202-2 This research was supported by the National Research Foundation of Korea (NRF-2019R1F1A1061368) and the World Institute of Kimchi (KE2202-2). The funders had no role in study design, data collection and analysis, decision to publish, or preparation of the manuscript.

==============================
Background

Ultramicrobacteria (UMB), also known as ultra-small bacteria, are tiny bacteria with a size less than 0.1 µm3. They have a high surface-to-volume ratio and are found in various ecosystems, including the human body. UMB can be classified into two types: one formed through cell contraction and the other that maintains a small size. The ultra-small microbiome (USM), which may contain UMB, includes all bacteria less than 0.2 µm in size and is difficult to detect with current methods. However, it poses a potential threat to food hygiene, as it can pass through sterilization filters and exist in a viable but non-culturable (VBNC) state. The data on the USM of foods is limited. Some bacteria, including pathogenic species, are capable of forming UMB under harsh conditions, making it difficult to detect them through conventional culture techniques.

Methods

The study described above focused on exploring the diversity of USM in fermented cabbage samples from three different countries (South Korea, China, and Germany). The samples of fermented cabbage (kimchi, suancai, and sauerkraut) were purchased and stored in chilled conditions at approximately 4 °C until filtration. The filtration process involved two steps of tangential flow filtration (TFF) using TFF cartridges with different pore sizes (0.2 µm and 100 kDa) to separate normal size bacteria (NM) and USM. The USM and NM isolated via TFF were stored in a refrigerator at 4 °C until DNA extraction. The extracted DNA was then amplified using PCR and the full-length 16S rRNA gene was sequenced using single-molecule-real-time (SMRT) sequencing. The transmission electron microscope (TEM) was used to confirm the presence of microorganisms in the USM of fermented cabbage samples.

Results

To the best of our knowledge, this is the first study to identify the differences between USM and NM in fermented cabbages. Although the size of the USM (average 2,171,621 bp) was smaller than that of the NM (average 15,727,282 bp), diversity in USM (average H′ = 1.32) was not lower than that in NM (average H′ = 1.22). In addition, some members in USM probably underwent cell shrinkage due to unfavorable environments, while others maintained their size. Major pathogens were not detected in the USM in fermented cabbages. Nevertheless, several potentially suspicious strains (genera Cellulomonas and Ralstonia) were detected. Our method can be used to screen food materials for the presence of USM undetectable via conventional methods. USM and NM were efficiently separated using tangential flow filtration and analyzed via single-molecule real-time sequencing. The USM of fermented vegetables exhibited differences in size, diversity, and composition compared with the conventional microbiome. This study could provide new insights into the ultra-small ecosystem in fermented foods, including fermented cabbages.

Introduction

Ultramicrobacteria (UMB) are less than 0.1 µm3 (or less than 0.3 µm) in diameter; the name UMB was first used by Torrella & Morita (1981) to describe extremely small bacteria. UMB are also referred to as ultra-small bacteria, nanobacteria, nano-organisms, dwarf cells, ultramicro cells, nano-sized microorganisms, filterable bacteria, small low nucleic acid-content bacteria, and nanobes (Velimirov, 2001; Duda et al., 2012; Ghuneim et al., 2018; Proctor et al., 2018). The small cell size of UMB provides a larger surface-to-volume ratio, thereby enabling the efficient absorption of nutrients in an oligotrophic environment (Giovannoni et al., 2005; Duda et al., 2012) and protecting against grazing pressure (Miteva & Brenchley, 2005; Williams et al., 2009). Therefore, UMB exist in various ecosystems, such as subterranean bedrock, soil, ocean, and river (Ghuneim et al., 2018), and the human body (Kajander & Ciftçioglu, 1998; He et al., 2015). While there is no official classification of UMB, they are classified into two types based on the effect of environmental factors on their morphology. The first type is UMB formed via cell contraction due to intrinsic and extrinsic factors, such as lack of nutrition or extremely harsh environments (Velimirov, 2001; Panikov, 2005), while the second type maintains a small size regardless of these factors and include strains different from existing taxa (Duda et al., 2012; Ghuneim et al., 2018). In this study, an ecosystem of bacteria less than 0.2 µm in size was confirmed; hence, filtrable bacteria including UMB and spores could be included. Therefore, we defined the ecosystem of all bacteria less than 0.2 µm in size as the ultra-small microbiome (USM). If a lethal pathogen is transformed into a UMB state, it is generally considered in a viable but non-culturable (VBNC) state (Kaprelyants, Gottschal & Kell, 1993). USM cannot be detected using the currently available culture-dependent methods and can therefore pass through sterilization filters with a pore size of ≤ 0.2 µm (Nakai, 2020). Therefore, USM remains a potential threat to food hygiene (Lee et al., 2022a).

Cabbages have various health benefits (Witzel, Kurina & Artemyeva, 2021). In particular, fermented cabbage can treat scurvy (Raak et al., 2014). A recent study has suggested that sulforaphane and lactic acid bacteria from fermented cabbage may help to lower the mortality rate of COVID-19 infections (Bousquet et al., 2021). Fermented cabbage is a popular item in Europe and North America, where it is processed and consumed as sauerkraut. In Asia, the type of cabbage preferred differs from that in Europe and is processed and consumed under the name “kimchi” in Korea and “suancai” in China. In El Salvador, cabbage is processed and consumed as “curtido” (Lee et al., 2022a).

To confirm the presence of USM in cabbage, this study investigated various types of fermented cabbage consumed in different regions and compared their microbial and ultramicrobial communities. Because fermented cabbage can greatly vary depending on the region, manufacturing method, and cultural tradition, studying only one type of fermented cabbage would not be representative. Therefore, this study selected samples from kimchi, sauerkraut, and suancai, which are representative of fermented cabbage in other regions, to ensure a comprehensive analysis. In addition, this study employed a novel strategy by combining tangential flow filtration (TFF) and single-molecule-real-time (SMRT) sequencing to differentially detect and sequence the 16S rRNA genes extracted from the ultra-small microbial communities.

We aimed to distinguish between normal size bacteria (NM) and ultra-small bacteria in fermented vegetables and identify the bacterial species with high resolution. Moreover, we attempted to understand the USM in food and identify potential pathogens, such as VBNC or persister states, which maximize the survival rate. To the best of our knowledge, this study is the first to examine the presence of USM in fermented foods, such as fermented cabbage, providing new insights into the potential risks associated with their presence.

Materials & Methods

Fermented vegetable samples

To explore the diversity of USM in fermented cabbage, kimchi, a fermented cabbage made in South Korea; suancai, a fermented cabbage made in China; and sauerkraut, a fermented cabbage made in Germany, were purchased (8 kg of kimchi, 4 kg of sauerkraut, and 5 kg of suancai) from an online market in June 2019. Moreover, the expiration date of kimchi, sauerkraut, and suancai from the date of manufacture was 1, 36, and 18 months, respectively. Both kimchi and suancai samples were non-sterile, while sauerkraut was sterile. Notably, suancai contained a preservative (potassium sorbate). Kimchi was made from kimchi cabbage (Brassica rapa subsp. pekinensis), suancai was made from Chinese cabbage (B. rapa subsp. pekinensis), and sauerkraut was made from white cabbage (Brassica oleracea var. capitata). B. rapa subsp. pekinensis, produced in Korea and bred for kimchi production, is called kimchi cabbage in order to be easily distinguished from B. rapa subsp. pekinensis, which is used to produce suancai (CAC, 2013). In addition, the fermented cabbages used in this study were not artificially produced in a laboratory, but were representative commercial products commonly consumed by the general public. The detailed ingredients of each fermented cabbage are shown in Fig. S1. The fermented cabbage samples were stored in chilled conditions at approximately 4 °C until filtration (Lee et al., 2022a).

Pre-filtration

Pre-filtration was performed on the broths of samples to facilitate TFF using a polypropylene capsule filter (GVS Filter Technology, Morecambe, UK) with a pore size of 10 µm. A vacuum pump (2546C-10; Welch Materials Inc., Shanghai, China) was used to aid the filtration process by reducing the pressure during the pre-filtration step. Before pre-filtration, the polypropylene capsule filter and tubing were sterilized with a solution of sodium hypochlorite 0.1% (v/v) (Lee et al., 2022a).

TFF

TFF was performed using a TFF system (Cogent µScale TFF System; Millipore, Sigma-Aldrich, St. Louis, IL, USA) in two phases. In the first phase, a TFF cartridge (Pellicon 2 Mini Cassette, Media: Durapore 0.22 µm; Millipore, Sigma-Aldrich) was used to cut off particles > 0.2 µm for the isolation of normal size bacteria as well as to remove macromolecules, including plasmid DNA. In addition, because UMB can pass through a 0.2-µm pore size filter due to its small size (Cavicchioli & Ostrowski, 2003; Duda et al., 2012; Nakai, 2020), a 0.2-µm pore size filter was used in many previous studies to identify UMB present in various environments (Silbaq, 2009; Loveland-Curtze, Miteva & Brenchley, 2010; Sahin et al., 2010; Maejima et al., 2018). Therefore, we set the pore size of the filter separating USM from NM to 0.2 µm to confirm USM based on UMB. Although USM encompasses endospores and OMVs including UMB, separation standard for the microbiome was set as UMB. In the second phase, a TFF cartridge (Pellicon 2 Mini Cassette, Media: Biomax 100 kDa; Millipore, Sigma-Aldrich) with a molecular weight cut-off (MWCO) of 100 kDa was used to remove small molecules below the USM size. Six samples, including (1) normal microbiome (NM) > 0.2 µm from kimchi (Kimchi_NM), (2) USM below 0.2 µm from kimchi (Kimchi_USM), (3) NM > 0.2 µm from sauerkraut (Sauerkraut_NM), (4) USM < 0.2 µm from sauerkraut (Sauerkraut_USM), (5) NM > 0.2 µm from suancai (Suancai_NM), and (6) USM < 0.2 µm from suancai (Suancai_USM), were subjected to further evaluation.

The samples were concentrated to approximately 25–50 fold via TFF followed by TFF system sterilization by recirculation with 0.1% (v/v) sodium hypochlorite for 30 min and cleaning by recirculation with sterilized ultrapure water for 2 h. The detailed specifications of the two phases of TFF are shown in Fig. S2. Additionally, Cai and colleagues (Cai et al., 2015) used TFF to efficiently separate bacteria and viruses from the marine environment and found that the adsorption of bacterial cells in a filter made of polyvinylidene fluoride (PVDF) was low (Cai et al., 2015). Based on their results, we used a cassette filter based on PVDF (Durapore 0.22 µm) for TFF to separate the USM < 0.2 µm and the NM > 0.2 µm. The USM and NM isolated via TFF were stored in a refrigerator at 4 °C until DNA extraction (Lee et al., 2022a).

DNA extraction and PCR amplification

Nucleic acids were extracted from 5 to 10 mL of concentrated NM and USM sample solutions using a DNeasy PowerSoil kit (Qiagen, Hilden, Germany) and quantified using the Quant-IT PicoGreen assay kit (Invitrogen, Thermo Fisher Scientific, Waltham, MA, USA) following the manufacturer’s instructions. Libraries were prepared via PCR amplification using the PacBio RS II. The nucleic acids were amplified with a primer set (27F, 5′-AGRGTTYGATYMTGGCTCAG-3′; 1492R, 5′-GGTTACCTTGTTACGACTT-3′) for the full-length 16S rRNA gene. The PCR conditions were as follows: initial denaturation at 94 °C for 5 min, followed by 35 cycles of denaturation at 94 °C for 30 s, annealing at 53 °C for 30 s, extension at 72 °C for 90 s, and a final extension at 72 °C for 5 min. Purification of the PCR amplicons was carried out using AMPure beads (Agencourt Bioscience, Beverly, MA, USA). To verify the amount and size of PCR products, fluorescence was measured using the Quant-IT PicoGreen assay kit (Thermo Fisher Scientific), and the template size distribution was measured using an Agilent DNA 12000 kit (Agilent Technologies, Santa Clara, CA, USA). Pooled amplicons were used for library preparation PacBio Sequel sequencing. A library was prepared using the PacBio DNA template prep kit 1.0 (Pacific Biosciences, Menlo Park, CA, USA). The PacBio DNA sequencing kit 4.0 and 8 SMRT cells (Pacific Biosciences) were used for sequencing (Lee et al., 2022a).

SMRT sequencing

SMRT sequencing was performed using a PacBio RSII system (Pacific Biosciences) according to the manufacturer’s instructions. In brief, 10-h movies were captured for each SMRT cell (Pacific Biosciences). Subsequent steps were carried out based on the PacBio Sample Net-Shared Protocol (https://www.pacb.com/). Circular consensus sequencing (CCS) reads, such as raw sequence reads, were processed using the SMRT analysis software (version 2.3; Pacific Biosciences). Short CCS reads and those with zero quality bases, considered as sequencing errors, were removed (Lee et al., 2022a).

Taxonomic and statistical analysis

Taxonomic analysis of the CCS reads was performed using the MG-RAST server (Meyer et al., 2008) with the SILVA SSU database (Quast et al., 2013). Then, in the pipeline version 4.0.3, the pipeline options were set as follows: assembled, dereplication, screening, length filtering, length filter deviation multiplicator, ambiguous base filtering, and maximum ambiguous base pairs were set to ‘yes,’ ‘no,’ ‘E. coli NCBI st. 536′, ‘2.00′, ‘no’, and ‘5′, respectively. Moreover, the e-value, percent identity, minimal alignment length, and minimal abundance values were set to ‘5′, ‘90′, ‘15′, and ‘1′, respectively. Additionally, ‘representative hit’ was selected. Statistical analyses were performed using MicrobiomeAnalyst (Dhariwal et al., 2017). Data normalization for each sample was performed for total sum scaling. Abundance profiling was presented as a stacked bar chart by calculating the percentage abundance, and less than 10 taxa were omitted. Species richness based on the alpha diversity of samples was determined via rarefaction curve analysis (McMurdie & Holmes, 2013), and the diversity of operational taxonomic units (OTUs) was indicated by the Shannon index. In addition, the evenness of OTUs was indicated by the pielou index (Jost, 2010). A heat tree was constructed using the non-parametric Wilcoxon rank sum test and was used to statistically quantify the hierarchical structure of taxonomic classification (Foster, Sharpton & Grünwald, 2017). The distance method and principal coordinate analysis (PCoA) was used for determining the beta diversity of samples and were set to unweighted UniFrac distance and permutational multivariate analysis of variance (PERMANOVA), respectively.

In addition, the distance measure and clustering algorithm of the hierarchical clustering algorithm (HCA) were set to the Bray–Cutis index and Ward, respectively. A heat tree was used to compare and sum the microbial communities and ultramicrobial communities for each sample. As a high-dimensional data analysis performed using a supervised machine learning algorithm, random forest classification analysis was conducted to identify the variability of strains in samples (Liaw & Wiener, 2002; Lee et al., 2022a).

Transmission electron microscopy (TEM)

For TEM observations via negative staining, droplets of the samples were mounted on a carbon support film on a 150-mesh nickel grid, stained with 4% uranyl acetate for 10 min and 0.4% lead citrate for 6 min, washed three times with deionized water, and air-dried. In addition, the samples were prepared in the form of ultrathin sections. To this end, samples were fixed by the addition of glutaraldehyde and paraformaldehyde adjusted to 2% in 0.05 M phosphate buffer (pH 7.2) and then incubated at room temperature (15–25 °C) for 4.5 h under vacuum. The fixed samples were washed three times for 15 min each with 0.05 M phosphate buffer at pH 7.2. The washed samples were post-fixed with osmium tetroxide adjusted to 1% in 0.05 M phosphate buffer (pH 7.2) at room temperature for 1 h. The post-fixed samples were washed three times for 15 min each with 0.05 M phosphate buffer at pH 7.2 and then dehydrated by passing through an ethanol gradient from 50 to 100%. After dehydration, the samples were precipitated to resin (LR white resin; EMS, Hatfield, PA, USA), placed in a disposable mold and embedded for 24 h at 60 °C. After the sample was hardened, ultrathin sections were prepared using an ultramicrotome equipped with a diamond knife and stained with 4% uranyl acetate for 10 min and 0.4% lead citrate for 6 min to complete sample preparation for TEM observation. The prepared samples were observed using a field-emission TEM (FE-TEM) (JEM-2100F; JEOL Ltd., Tokyo, Japan) at 200 kV accelerating voltage (Lee et al., 2022a).

Data availability

The sequencing reads of fermented cabbages were deposited to the NCBI under BioProject ID PRJNA684410, the metadata for each sample can be found in SRR13260135, SRR13260137, SRR13260138, SRR13260156, SRR13260177, SRR13260178.

Results

SMRT sequencing

The NM and USM in fermented cabbage were separated using TFF and analyzed via SMRT sequencing. SMRT sequencing read quality indices are shown in Table S1. Sequence read information for each sample is presented in Table 1. A total of 19,356 sequence reads from Kimchi_NM, 1,383 from Kimchi_USM, 1,208 from Sauerkraut_NM, 1,610 from Sauerkraut_USM, 11,446 from Suancai_NM, and 1,570 from Suancai_USM were generated. The total number of reads from NM was much greater than that of USM. Notably, the number of reads from the Sauerkraut_USM was slightly higher than that from the Sauerkraut_NM. Kimchi_NM had the greatest number of sequence bases at 28,690,199 bp, while Sauerkraut_NM had the lowest at 1,741,058 bp. The average read lengths of the samples ranged from 1,425 to 1,482 bp, and were almost identical between groups (Lee et al., 2022a).

Table 1 States of sequence reads for each sample after trimming.

Sample name	Read counts	Total read bases (bp)	Average read length (bp)	
Kimchi_NM	19,356	28,690,199	1,482 ± 278a	
Kimchi_USM	1,383	1,983,758	1,434 ± 271	
Sauerkraut_NM	1,208	1,741,058	1,441 ± 288	
Sauerkraut_USM	1,610	2,293,503	1,482 ± 258	
Suancai_NM	11,446	16,750,589	1,463 ± 296	
Suancai_USM	1,570	2,237,604	1,425 ± 246	
Notes.

Kimchi_NM normal microbiome > 0.2 µm in size from kimchi

Kimchi_USM ultra-small microbiome < 0.2 µm in size from kimchi

Sauerfkraut_NM normal microbiome > 0.2 µm in size from sauerkraut

Sauerkraut_USM ultra-small microbiome < 0.2 µm in size from sauerkraut

Suancai_NM normal microbiome > 0.2 µm in size from suancai

Suancai_USM ultra-small microbiome < 0.2 µm in size from suancai

a Standard deviation for the average read length.

Taxonomic and statistical analyses

Community richness in the samples was expressed using the rarefaction curve (Fig. 1A), while alpha diversity was expressed using the Shannon index (Fig. 1B). Each sample in the rarefaction curve plateaued. Suancai_NM had the highest richness (OTUs of 84), while Sauerkraut_NM had the lowest (OTUs of 11). However, generally, the OTUs in Kimchi_NM and Suancai_NM were more abundant than those in the Kimchi_USM, Sauerkraut_USM, and Suancai_USM, but not than that in Sauerkraut_NM. Since the rarefaction curve of Kimchi_NM was gently inclined, richness was high, while diversity was low. Based on the Shannon index, Kimchi_NM had the lowest diversity at 0.26, while Kimchi_USM had 1.62. Comparing fermented cabbage types, Sauerkraut_NM had a diversity of 1.45, Sauerkraut_USM had 1.13, Suancai_NM displayed the highest diversity at 1.95, and Suancai_USM had 1.23. Suancai_NM had the highest microbial diversity, whereas Kimchi_NM had the lowest. The alpha diversity of suancai was higher than that of kimchi and sauerkraut when diversity was assessed between fermented cabbage types. Further, USM had higher alpha diversity on an average when diversity was assessed based on community type and regardless of the sample type. Based on the Pielou index, Kimchi_NM had the lowest evenness at 0.03, while Kimchi_USM displayed the highest evenness at 0.43. Comparing fermented cabbage types, Sauerkraut_NM had an evenness index of 0.31, Sauerkraut_USM had 0.23, Suancai_NM had 0.26, and Suancai_USM had 0.26. In terms of evenness among fermented cabbage types, sauerkraut and suancai showed relatively similar values. While assessing evenness based on community type and regardless of sample type, USM samples generally demonstrated higher evenness compared with NM samples, with the exception of sauerkraut, where NM samples had higher evenness than USM samples (Lee et al., 2022a).

Figure 1 Rarefaction curve (A) and Shannon index (B) of microbial and ultramicrobial communities detected in fermented cabbages.

Relationship between number of operational taxonomic units (OTUs) and sequences was applied to a rarefaction curve, and each sample was plateaued. Shannon index was expressed for each sample (left), cabbage type (middle), and community (right).

The NM and USM in kimchi and suancai are shown in Fig. 2A and Fig. S3. At the phylum level, Firmicutes were dominant in Kimchi_NM and Suancai_NM. In the former, Firmicutes accounted for 100% of the microbial community. Actinobacteria dominated Kimchi_USM, and uncultured bacteria dominated Sauerkraut_USM as well as Suancai_USM. At the species level, Weissella koreensis was dominant at 94% in Kimchi_NM, while Weissella cibaria was also present, but in minor quantities. Cellulomonas uda and Cupriavidus pauculus were predominant in Sauerkraut_NM, at 32% and 26%, respectively. However, in Sauerkraut_NM, uncultured bacteria accounted for a significant proportion (32%). Lactobacillus acetotolerans was dominant (53%) in Suancai_NM. Lactobacillus similis was also predominant in Suancai_NM, accounting for 11%. Cellulomonas biazotea dominated at 42% in Kimchi_USM. In particular, candidate division TM7 single-cell isolate TM7a (TM7a), known as Saccharibacteria, was predominant at 35%. Cellulomonas uda predominated in Sauerkraut_USM at 36%. However, uncultured soil bacteria dominated Sauerkraut_USM at 54% and Suancai_USM at 48% (Lee et al., 2022a).

Figure 2 Relative abundance profiling (A), principal coordinate analysis (PCoA) plot (B), and hierarchical cluster analysis (HCA) dendrogram (C) reflected the species-level abundance.

OTUs with an abundance below 10 as determined via relative abundance profiling were expressed as others. The statistical significance of the clustering pattern in the PCoA plot was evaluated through permutational ANOVA (PERMANOVA). The distance measure and clustering algorithm of HCA were applied to the Bray–Cutis index and Ward, respectively.

The OTU values indicated that the species richness of NM is generally higher than that of USM (Fig. 1). In particular, Suancai_NM exhibited the highest abundance and alpha diversity, as indicated by significantly higher OTU and Shannon index values of the microbiome compared to that of the other samples. However, the Pielou index value of NM was 0.26, which was relatively low compared with other samples, and was the same as that of USM. Therefore, Suancai’s USM and NM were almost equal in evenness, and evenness was inferior to that of Sauerkraut_NM with a Pielou index value of 0.31. For Kimchi_NM, the OTU value was relatively higher than that for other samples, however, its Shannon index value was the lowest. In addition, the rarefaction curve of Kimchi_NM only slightly increased, suggesting low diversity. The Shannon index was lower than the OTU value because Weissella koreensis showed > 94% dominance. In contrast, L. acetotolerans showed > 53% dominance in Suancai_NM, while other species showed minor dominance, between 1 and 11%. Therefore, the Shannon index of Suancai_NM was estimated to be the highest. In addition, The OTU values of USM were relatively low, with the one for Kimchi_USM being the lowest. However, Shannon index of Kimchi_USM was the second highest, > 1.6, possibly due to the even distribution of species. As proof of this, the Pielou index value of Kimchi_USM was 0.43, indicating higher evenness than other samples. While the difference in alpha diversity values between the NM and USM of kimchi and suancai was high, the difference in alpha diversity value between the NM and USM of sauerkraut was small. Since sterilization was performed during the manufacturing process of sauerkraut, most of the normal microorganisms did not survive. Therefore, the difference in alpha diversity between Sauerkraut_NM and Sauerkraut_USM was not expected to be high. The species richness of USM was low, yet the diversity was higher than that of NM. The microbial distribution of the fraction that passed through the 0.2-µm filter, which is sterile, was more diverse than expected (Nakai, 2020).

The beta diversity of NM and USM in fermented cabbage samples was compared via PCoA (Fig. 2B) and HCA (Fig. 2C). The plot and dendrogram showed that Suancai_NM and Kimchi_USM were closely related as also observed for Sauerkraut_NM, Sauerkraut_USM, and Suancai_USM. In addition, PCoA indicated that Kimchi_NM was separated from the other samples. However, Suancai_NM, Kimchi_USM, and Kimchi_NM were grouped in the HCA dendrogram, as were Suancai_USM, Sauerkraut_NM, and Sauerkraut_USM (Lee et al., 2022a).

A heat tree was used to compare and sum the NM and USM at the genus level for each fermented cabbage (Fig. 3). The sum of NM and USM per fermented cabbage type is shown in Figs. 3A–3C. Obtaining this sum was equivalent to combining the NM and USM of each fermented cabbage in the bar chart of Fig. 2A. Figures 3D–3F compare the NM and USM in each fermented cabbage. In kimchi, several genera belonging to the phylum Firmicutes had a relatively high ratio in the NM compared to the USM (Fig. 3D). Unlike in kimchi, in sauerkraut, several genera belonging to the phylum Proteobacteria had a relatively higher ratio in the NM compared to that in the USM (Fig. 3E). Suancai was similar to kimchi as several genera belonging to the phylum Firmicutes had a relatively higher ratio in the NM compared to that in the USM (Fig. 3F) (Lee et al., 2022a).

Figure 3 A heat tree used to compare and sum the microbial and ultramicrobial communities for each sample at the genus level.

(A–C) The total microbiome of kimchi, sauerkraut, and suancai, respectively; (D–F) comparison between the NM and USM of kimchi, sauerkraut, and suancai, respectively.

The random forest algorithm was used to confirm the top 15 OTUs with large variability between the microbiomes (Fig. 4). The mean reduced accuracy represents the accuracy that the microbiome loses by excluding each variable. Thus, the lower the accuracy, the more important the variable species is for a successful classification. Species are displayed in descending order of importance, that is, the higher the mean reduction accuracy, the higher the importance of the species in the microbiome (Martinez-Taboada & Redondo, 2020). Since the mean decrease accuracy of uncultured bacteria was the highest (0.0279), the microbiome was likely to be divided based on the content of these strains. The mean decrease in accuracy of uncultured Azospira sp. and Lactobacillus paracollinoides was 0.0231 and 0.0230, respectively, which were the second and third highest, respectively (Lee et al., 2022a).

Figure 4 Random forest classification analysis.

Analysis confirmed the top 15 OTUs with greatest variability between communities. The mean reduced accuracy represents the accuracy that communities lose by excluding each variable. The lower the accuracy, more important the variable species for a successful classification.

Morphological observations via TEM

TEM was used to confirm the presence of microorganisms in the USM of fermented cabbage samples (Fig. 5 and Fig. S4). Most of the USM were cocci with both outer and inner membranes observed in the USM isolated from all fermented cabbages. In addition, the periplasmic space between the outer and inner membranes was observed (Figs. 5C, 5G and 5K). The size of the microorganism in USM was approximately 100–200 nm. In addition, USM isolated from fermented cabbages had multiplied via dichotomy (Fig. 5B, Figs. S4H and S4K) (Lee et al., 2022a).

Figure 5 Transmission electron micrographs for ultramicrobial communities after ultra-section of fermented cabbages.

Transmission electron micrographs of the ultramicrobial community of a size < 0.2 µm in (A–D) kimchi (Kimchi_USM), (E–H) sauerkraut (Sauerkraut_USM), and (I–L) suancai (Suancai_USM). DT, dichotomy; IM, inner membrane; PS, periplasmic space: OM, outer membrane.

Discussion

In the present study, we identified the NM and USM in different fermented cabbages via TFF and SMRT to characterize and compare USM between cabbage types. TFF has been used to concentrate various microorganisms in water and is an excellent technique for their separation or removal (Cai et al., 2015). TFF has shown 11–98% recovery for plankton viruses, smaller than 0.2 µm in size, from freshwater samples (Colombet et al., 2007). Therefore, in the present study, we employed TFF instead of conventional normal flow filtration (NFF) to separate and concentrate NM and USM from kimchi, sauerkraut, and suancai. TFF is more efficient than NFF since it can filter more liquid phase by smoothly removing the filter cake. In addition, while there were fewer sequence reads for USM than those for NM (Table 1), they might not have been obtained if not enriched via TFF (Lee et al., 2022a).

SMRT sequencing, a synthetic long-read sequencing technology based on full-length 16S rRNA, is widely used to study microbial communities in various environments, including food. It has been used to analyze microbial diversity in various samples (Pootakham et al., 2017; Jeong et al., 2021; Hui et al., 2022; Liang et al., 2022). SMRT sequencing is advantageous over general Illumina sequencing because it provides accurate classification at a taxonomic level using full-length 16S rRNA sequences instead of short amplicons (Mosher et al., 2014; Jeong et al., 2021); in this study, SMRT sequencing was used to confirm the microbial composition of fermented cabbage, including NM and USM samples.

Moreover, we categorized the sequences as operational taxonomic units (OTUs). Indeed, recent studies suggest that amplicon sequence variants (ASVs) offer a higher taxonomic resolution than OTUs, and additionally correct sequencing errors, thereby providing a more accurate representation of microbial diversity (Callahan, McMurdie & Holmes, 2017; Nearing et al., 2018; Chiarello et al., 2022). However, it is worth noting that no significant difference has been found between the results yielded by analyses of OTUs and ASVs (Glassman & Martiny, 2018). OTUs have long served as a simple and intuitive method of classification, and due to their widespread use over the years, a greater number of studies employ OTUs compared to ASVs. This prevalence allows for a more comprehensive comparison of our research with existing literature.

Weissella koreensis, a dominant bacterium in kimchi (Jung, Lee & Jeon, 2014), was first reported in kimchi in 2002 (Lee et al., 2002). Further, Weissella spp., including W. koreensis, are involved in kimchi fermentation (Cho et al., 2006). In line with previous reports, this study also showed that W. koreensis was highly prevalent in Kimchi_NM. However, in Kimchi_USM, C. biazotea, which is known to degrade cellulose (Rajoka & Malik, 1997), and TM7a, which is known to be parasitic on bacterial hosts (Marcy et al., 2007; Bor et al., 2019), were both detected (Lee et al., 2022a).

The microbial abundance of USM and NM in each sample was relatively normalized via total sum scaling, however, because the number of sequence reads from Kimchi_USM was over ten times less than that of Kimchi_NM, C. biazotea and TM7a do not represent the microbiome in general kimchi containing USM and NM. In addition, based on the sequence reads, the present ratio of USM to NM in kimchi was 7.1, which was the basis for the lower microbial abundance in USM compared with NM. Further, if the USM was not separated via TFF, C. biazotea and TM7a would not have been identified in kimchi. We reviewed the available literature on the microbiome of kimchi and found no reports on C. biazotea and TM7a in kimchi via culture-dependent or culture-independent methods (Cho et al., 2006; Park et al., 2012; Jung, Lee & Jeon, 2014; Lee et al., 2017; Maoloni et al., 2020; Park et al., 2020; Lee et al., 2022a).

The 16S rRNA gene is shared among all bacteria and utilizing this gene would significantly reduce the labor and cost of profiling the identity and abundance of microorganisms in various environments, regardless of the culture capacity (Hugenholtz et al., 2021). However, the 16S rRNA gene is not an optimal target, owing to the short read length of most commonly used sequencing platforms, such as Illumina, which limits the taxonomic resolution to families or genera (Earl et al., 2018; Jeong et al., 2021). However, if the 16S rRNA gene is almost entirely sequenced using SMRT sequencing, which can also analyze long sequences, taxonomic resolution can be improved. In a previous study, 60% of specific phyla, such as the phylum Microgenomates, were not detected via PCR using the 518F and 806R primer sets (Brown et al., 2016), and the low taxonomic resolution due to short sequencing limits the amount of data on microbial ecology (Lee et al., 2022a). Indeed, SMRT sequencing offers a significant advantage over Illumina sequencing in terms of taxonomic resolution because it covers the entire variable region (V1–V9) of the 16S rRNA gene (Mosher et al., 2014). In contrast, Illumina sequencing typically covers only a narrow region. This increased resolution allows for accurate identification and classification of microbial taxa, particularly for rare or hard-to-detect species, which may not be as effectively characterized using short-read sequencing platforms, such as Illumina. Therefore, the identification of C. biazotea and TM7a in kimchi may be attributed to the use of SMRT sequencing in the current study. In particular, TM7, also known as Saccharibacteria, have been reported in the oral cavity (Bor et al., 2019) and are known to be ultra-small (200–300 nm) and parasitic bacteria attached to the surface of host bacteria (Bor et al., 2016), which complicates their detection via conventional culture methods. Further, TM7 does not grow unless a special method of symbiosis is employed (Murugkar et al., 2020). The causal relationship observed herein remains unclear, and further studies are warranted to determine how TM7 is transmitted to humans. In addition to C. biazotea and TM7a, other bacteria were rarely found in Kimchi_USM. Therefore, most of the bacteria in the USM did not adapt to the kimchi environment or lost their survival competition to Weissella spp. (Lee et al., 2022a).

Sauerkraut_NM and Sauerkraut_USM were dominated by C. uda. Like C. biazotea, C. uda secretes cellulases. The genus Cellulomonas is abundant in soil (Robinson, Batt & Batt, 2014) and was found in Sauerkraut_NM and Sauerkraut_USM, probably derived from the soil where the raw cabbage had been planted. Furthermore, while the genus Cellulomonas is relatively rare, it may potentially be pathogenic because there have been instances of human infection (Salas et al., 2014). Hahn et al. (2003) discovered novel USM from the class Actinobacteria, belonging to the genus Cellulomonas, in five freshwater reservoirs. Although only Actinobacteria living in freshwater environments were mentioned, Cellulomonas in soil should also have several USM types. Cupriavidus pauculus (also known as Ralstonia paucula), detected only in Sauerkraut_NM, can pass ultrafiltration (Cuadrado et al., 2010), although it was not found in Sauerkraut_USM. In addition, C. pauculus may not occur in a USM, because it is a filterable bacterium (Lee et al., 2022a).

Suancai_NM was dominated by L. acetotolerans, which was first reported in fermented vinegar broth (Entani, Masai & Suzuki, 1986). Lactobacillus spp. are involved in suancai fermentation (Liu et al., 2019). However, L. acetotolerans was previously reported as abundant in pao cai, but not in suancai (Cao et al., 2017; Liu et al., 2019). These differences are likely attributable to variations in the manufacturing processes (e.g., temperature, salinity, and seasoning), and these distinctions can be observed in Fig. S1. Uncultured bacteria, including soil bacteria, were dominant in Suancai_USM, and these OTUs may not be included in the SILVA SSU database. Similarly, in Sauerkraut_NM and Sauerkraut_USM, uncultured soil bacteria and uncultured bacteria were also dominant, while random forest analysis indicated that their OTUs exhibited large variation among samples (Fig. 4) (Lee et al., 2022a).

Ralstonia spp. were detected in both Sauerkraut_USM and Suancai_USM at < 1%. However, Ralstonia spp., similar to C. pauculus, are filterable bacteria, hence, they are classified as USM but are most likely not UMB. In addition, since they can survive by attaching to the ultrapure water system (Kulakov et al., 2002), they may not be resident microorganisms in kimchi or suancai. Detected at < 1%, Ralstonia spp. do not present a major bias. Nevertheless, some species within these genera are pathogens, and may potentially serve as pathogenic agents; a thorough sterilization may be necessary when similar studies are conducted in the future.

Comparison of the microbial and ultramicrobial communities of kimchi, sauerkraut, and suancai via PCoA and HCA (Figs. 2B and 2C) showed differences between them. In addition, although each fermented cabbage had distinct microbial community (Figs. 3A–3C), the heat tree indicated differences between sample NM and USM (Figs. 3D–3F). While the main ingredients (Brassica rapa subsp. pekinensis) of kimchi and suancai are similar, differences in the manufacturing method, other ingredients, seasoning, and the surrounding environment might have contributed to the prevalence of different microbial and ultramicrobial communities. Since sauerkraut was sterilized in the manufacturing process, the difference between the microbial and ultramicrobial communities was not high and was very similar to Suancai_USM. In sauerkraut, the genes of both NM and USM were mixed with those of dead cells due to sterilization; the reason behind the microbial abundance not being high, even though the genes of the dead cells were mixed, is presumed to be that the genome of the dead bacteria was decomposed by the high temperature during the sterilization process (Eichmiller, Best & Sorensen, 2016; Tsuji et al., 2017; Lee et al., 2022a).

Although we successfully identified USM between 0.2 µm and 100 kDa in fermented vegetables, TFF and SMRT sequencing methodologies may lead to misrecognition of fragments of bacteria (Duda, Suzina & Boronin, 2020). Therefore, we sought to confirm the presence of ultra-small microorganisms in USM via TEM (Fig. 5 and Fig. S4). The ultra-small microorganism in Kimchi_USM, Sauerkraut_USM, and Suancai_USM were observed as coccoid types, 100–200 nm in size, with outer and inner membranes, as well as multiplication via dichotomy; therefore, the existence of ultra-small microorganism in the UMB of fermented cabbages was confirmed. The possibility that it is an outer membrane vesicle cannot be excluded (Cecil et al., 2019). However, bisection was observed; therefore, the possibility that it is an ultra-small microorganism cannot be ruled out (Lee et al., 2022a).

Rod shape bacteria such as genera Cellulomonas, Cupriavidus, and Lactobacillus were also detected in USM using SMRT, while only spherical shape bacteria were found in TEM. Since, as previously suggested, spherical shaped bacteria must be at least 250–300 nm in diameter to maintain the 250–300 proteins essential for life, the normal state may therefore be rod shape; however, it might have transformed into a spherical shape for survival (Ghuneim et al., 2018). Moreover, USM formation, even in nutrient-rich environments, was possibly attributed to the action of predators and selective pressures such as pH and drying (Simon et al., 2002; Pernthaler, 2017; Lee et al., 2022a).

In NM or USM extracted from fermented cabbages, major human pathogenic bacteria were not detected. However, although the number of OTUs and sequence reads were small, several taxonomic groups suspected of causing human diseases, such as Ralstonia were detected in USM, and a significant proportion of uncultured bacteria with or without pathogenicity were also detected. Although these bacteria were smaller than the general pore size of the sterilizing filter, which is 0.2 µm, and may not be harmful immediately, there is a concern that there may be potential risks when the fermented cabbages are further processed into sauces, juices, etc. Furthermore, based on a previous study (Lee et al., 2022b), it was suspected that USM may be in a VBNC state. Although USM was not starving due to the abundant nutrients in the fermented cabbages, it is possible that the dominance of lactic acid bacteria due to fermentation could have led to stress and difficulty in growth, resulting in conversion to ultra-small microorganisms in a VBNC state. VBNC bacteria can survive under severe stress conditions, such as starvation, low-temperature sterilization, and antibiotics, and some subgroups can survive without time-dependent changes in gene expression as a response to stressful stimuli (Chaveerach et al., 2003; Li et al., 2014; Ramamurthy et al., 2014; Ayrapetyan & Oliver, 2016). In addition, VBNC bacteria, similar to persistent bacteria, probabilistically can exist even in rapidly growing environments (Ayrapetyan et al., 2015; Gonçalves & De Carvalho, 2016; Orman et al., 2016). Therefore, the transition to a VBNC state is an important issue in food hygiene (Dong et al., 2020), and USM should also be considered as having potential risks. In addition, Li and colleagues reported that the use of acetic acid can inhibit the VBNC state of E. coli O157:H7 (Li et al., 2020). Thus, it is possible to reduce the potential risks of USM suspected of being in a VBNC state by adjusting the concentration of organic acids or using other methods.

Although this study was conducted rigorously, there remains the possibility of contamination from external microbial DNA during the sampling or DNA extraction process. Moreover, it may be challenging to consider the samples in this study as fully representative of all fermented cabbage varieties. However, this study holds significance because it is the first to investigate USM in nutrient-rich foods such as fermented cabbages, shedding light on the potential risks associated with their presence. Therefore, we believe that this study will serve as a solid foundation for future studies on fermented foods and USM fostering further advancements in the field.

Conclusions

In conclusion, our findings revealed that USM in cabbage differed from NM, and although major pathogens were not detected within USM, several potentially concerning strains, such as those from the genera Cellulomonas and Ralstonia, were detected. This suggests that poor sanitization in the manufacturing environment could lead to the presence of ultra-small microorganisms in fermented food products, posing a risk to consumer health. Overall, our study offers new insights into the USM in food, specifically in fermented cabbage, and underscores the importance of proper sanitization during production.

Supplemental Information

Figure S1 Detailed ingredients of each fermented cabbage

The red boxes represent the ingredients. (A) kimchi, (B) sauerkraut, (C) suancai.

Click here for additional data file.

Figure S2 Diagram of the tangential flow filtration (TFF) process

(A) Tank. The solution to be filtered is added. The unfiltered solution is then collected and concentrated via TFF. (B) Peristaltic feed pump. The solution is pumped through the filter membrane by a peristaltic feed pump. (C) TFF cartridge and holder, and where TFF filtration takes place. Filtration was performed with a 0.22 µm pore size or a 100 K molecular weight cut-off (MWCO) filter cartridge installed. (D) Filtrate collection container where the filtered solution is collected.

Click here for additional data file.

Figure S3 Relative abundance profiling reflected abundances at the phylum, class, order, family, and genus levels

Click here for additional data file.

Figure S4 Transmission electron micrographs after negative staining for ultramicrobial communities of fermented vegetables

Transmission electron micrographs of the ultramicrobial community of a size < 0.2 µm in (A–D) kimchi (Kimchi_USM), (E–H) sauerkraut (Sauerkraut_USM), and (I–L) suancai (Suancai_USM).

Click here for additional data file.

Table S1 SMRT sequencing read quality indices

Click here for additional data file.

We thank Macrogen (South Korea) for their assistance with the SMRT sequencing.

Additional Information and Declarations

Competing Interests

Author Contributions

DNA Deposition

Data Availability

The authors declare there are no competing interests.

Hae-Won Lee conceived and designed the experiments, performed the experiments, analyzed the data, prepared figures and/or tables, authored or reviewed drafts of the article, and approved the final draft.

So-Ra Yoon analyzed the data, prepared figures and/or tables, and approved the final draft.

Yun-Mi Dang performed the experiments, authored or reviewed drafts of the article, and approved the final draft.

Miran Kang analyzed the data, authored or reviewed drafts of the article, and approved the final draft.

Kwangho Lee performed the experiments, prepared figures and/or tables, and approved the final draft.

Ji-Hyung Ha conceived and designed the experiments, analyzed the data, authored or reviewed drafts of the article, and approved the final draft.

Jin-Woo Bae conceived and designed the experiments, authored or reviewed drafts of the article, and approved the final draft.

The following information was supplied regarding the deposition of DNA sequences:

The sequencing reads of fermented cabbages are available at NCBI: PRJNA684410.

The following information was supplied regarding data availability:

Data is available at NCBI GenBank: SRX9690674, SRX9690676, SRX9690677, SRX9690695, SRX9690716, SRX9690717.

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
