# Peer review of "Presence of an ultra-small microbiome in fermented cabbages"

_PeerJ, doi:10.7717/peerj.15680_

## Round 0.1 · original submission · Major Revisions

Overall, the study on the diversity of ultra-small microbiome (USM) in fermented cabbage samples using tangential flow filtration (TFF) and single-molecule-real-time (SMRT) sequencing is an interesting contribution to the field. However, there are several points that could be further elaborated or discussed:

- Lack of information on the sample size and representativeness of the study: The article does not provide details on the number of samples analyzed or how representative they are of the different regions and types of fermented cabbage. Without this information, it is difficult to assess the generalizability of the findings.

- Limited information on the filtration process: The article briefly mentions that TFF was used to separate normal size bacteria (NM) and USM, but it does not provide sufficient details on the process or the rationale for selecting the pore sizes of the cartridges used. It would be helpful to provide more information on this aspect of the methodology to enable other researchers to replicate the study.

- Lack of validation of the SMRT sequencing results: While SMRT sequencing is a powerful tool for identifying bacterial species, it is not clear if the results were validated using other methods, such as quantitative PCR or 16S rRNA gene sequencing. Validation of the SMRT sequencing results would increase the confidence in the findings.

- Limited discussion on the potential implications of the findings: The article mentions that the USM in fermented vegetables differed from the conventional microbiome, but it does not elaborate on the potential implications of this difference. For example, are there any health risks associated with the presence of USM in fermented vegetables, and how can this knowledge be used to improve food safety?

- Lack of discussion on the limitations of the study: The article does not discuss any limitations of the study, such as the possibility of contamination during the sampling or DNA extraction process. It would be helpful to acknowledge and discuss any potential limitations of the study to enable readers to interpret the findings appropriately.\

·

Basic reporting

this article deals with an interesting topic about ultra small bacteria and microbiomes, it has relevant results. However it has some issues related to unprofessional figures and tables, some of the literature references are so old (i.e 1918), and some ideas and affirmations need to be improved before its publication at PeerJ

Detailed comments and suggestions are in the attached file. I will be honored to review the corrected version of this manuscript.

Experimental design

Their experimental design is a quite robust, However authors do not have enough information about their metadata. Moreover, the bioinformatics methods used in their manuscript are not described with sufficient detail to replicate their pipelines.

Validity of the findings

The benefit to literature is clear, however the authors need to reinforce their conclusions because actually are not so clear. Also they need to link their findings with their research question

·

Basic reporting

Dear Authors
The manuscript entitled “Presence of an ultra-small microbiome in fermented cabbages” ideas is very innovative and significant to food industries.
But your experiment is not looking logical and significant. The first question arises: how can you separate the fermenting material, NM and UMB? You mentioned in lines no 41-43 about separation through various filters but it is impossible to separate only UMB. The resultant may contain various components of fermenting substrate and by using pressure may be modified shape and size of NM. After separation, you isolated the DNA and 16S rRNA sequencing and you reported various microbes which are important for fermentation such as LABs so how can you say that these are UMB and how can report some others are pathogenic without pathogenic study? Hence, these types of studies create problems to food industries.

Experimental design

Not appropriate

Validity of the findings

The experiment itself not correct so not matter of findings.

Additional comments

No

·

Basic reporting

The manuscript titled “Presence of an ultra-small microbiome in fermented cabbages’’ represent a significant contribution to scholarly research and is good for publication but needs minor revisions
There are lots of questions that I highlighted in the reviewed text that need to be addressed before acceptance for publication.
1. Abstract: The information in the results section of the abstract is not clearly stated, values of the sizes USM and NM of various cabbage samples screened should be stated . The diversity index value should also be stated
2. In introduction; The clear objectives of the study should be stated. Lines 90- 99 should be rephrased to reflect the clear objective of the study
4. Materials and methods should be detailed. I highlighted the parts that need clarification in
the text. See the attached reviewed manuscript -
5. Results: The values of some parts of the results were not stated eg, alpha diversity index values , Shannon index value
Figures 1- 3 are not clear enough, a clearer version should be provided
Discussion. Some part of the discussion are basically the detail interpretation of the results – see lines 287-303 -should be moved to the appropriate results section
Conclusion: The conclusion did not reflect the summary of the study, thus should be rewritten. The basic findings of the study should be concisely stated in the conclusion section.
Other comments are found in the reviewed text.

Experimental design

No comment

Validity of the findings

No comment

Additional comments

Authors should ensure that they respond to all the questions I raised on the reviewed documents

---

## Round 0.2 · Minor Revisions

The reviewers found your manuscript suitable for consideration for publication in PeerJ. Before publication, however, several questions raised by reviewer 1 need to be addressed.

·

Basic reporting

Authors have improved their manuscript in all suggested aspects, indeed is more reliable and clear to understand their contribution to the field.

Actually, I think that the authors need to prepare any proof correction of their text, because with all the reviewers suggestions there are certain sections that have double or triple spaces and on the other hand there are missing some punctuation marks.

Moreover, the authors improved their figures to made them more professional.

Experimental design

This manuscript has a well defined research question and its contribution is clear, however, authors need to emphasize why use OTUs instead of ASVs, I read that they used mg-rast and mg-rast prepare OTUs, but that is not a scientific response.

I recommend to the authors to revise this articles to prepare their response in a more scientific and professional way.

Chiarello, Marlène, et al. "Ranking the biases: The choice of OTUs vs. ASVs in 16S rRNA amplicon data analysis has stronger effects on diversity measures than rarefaction and OTU identity threshold." PLoS One 17.2 (2022): e0264443.

Callahan, Benjamin J., Paul J. McMurdie, and Susan P. Holmes. "Exact sequence variants should replace operational taxonomic units in marker-gene data analysis." The ISME journal 11.12 (2017): 2639-2643.

Nearing JT, Douglas GM, Comeau AM, Langille MG. Denoising the Denoisers: an independent evaluation of microbiome sequence error-correction approaches. PeerJ. 2018 Aug 8;6:e5364.

Validity of the findings

This article has robust analysis and actually is statistically sound and controlled. Moreover, conclusions are well stated and linked with the original research question

Additional comments

The authors have done an excellent job to improve their manuscript so all that remains is to answer a couple of questions and check for typos

---

## Round 0.3 · accepted · Accept

The authors have revised the manuscript as per the reviewer suggestions. Now manuscript can be accepted in its current form